# Micropropagation, Genetic Fidelity and Phenolic Compound Production of *Rheum rhabarbarum* L.

**DOI:** 10.3390/plants9050656

**Published:** 2020-05-22

**Authors:** Doina Clapa, Orsolya Borsai, Monica Hârța, Victoriţa Bonta, Katalin Szabo, Vasile Coman, Otilia Bobiș

**Affiliations:** 1Institute of Advanced Horticulture Research of Transylvania, University of Agricultural Sciences and Veterinary Medicine Cluj-Napoca, Mănăștur st. 3-5, 400372 Cluj-Napoca, Romania; doina.clapa@usamvcluj.ro; 2Life Science Institute, University of Agricultural Sciences and Veterinary Medicine Cluj-Napoca, Mănăștur st. 3-5, 400372 Cluj-Napoca, Romania; monica.harta@usamvcluj.ro (M.H.); victorita.bonta@usamvcluj.ro (V.B.); katalin.szabo@usamvcluj.ro (K.S.); vasile.coman@usamvcluj.ro (V.C.); 3AgroTransilvania Cluster, Dezmir, Crișeni FN, 407039 Cluj, Romania

**Keywords:** growth regulators, genetic fidelity, rhubarb, rosmarinic acid, resveratrol, phenolic derivatives

## Abstract

An efficient micropropagation protocol for *Rheum rhabarbarum* L. was developed in this study. The in vitro rhubarb plants obtained in the multiplication stage (proliferation rate: 5.0 ± 0.5) were rooted in vitro (96% rooting percentage) and acclimatized *ex vitro* in floating perlite, with 90% acclimatization percentage. To assess the genetic fidelity between the mother plant and in vitro propagated plants, sequence-related amplified polymorphism (SRAP) markers were used. All banding profiles from the micropropagated plants were monomorphic and similar to those of the mother plant indicating 100% similarity. Regarding the polyphenolic profile, gallic, protocatechuic, p-hydroxybenzoic, vanillic, chlorogenic, caffeic, syringic, p-coumaric and ferulic acid were present in different amounts (2.3–2690.3 μg g^−1^ dry plant), according to the extracted matrix. Aglicons and glycosides of different classes of flavonoids were also identified. The rhizome extracts (both from in vitro and field grown plants) contained resveratrol, a stilbene compound with high antioxidant properties, ranging between 229.4 to 371.7 μg g^−1^ plant. Our results suggest that in vitro propagation of *Rheum rhabarbarum* L. represents a reliable alternative to obtain a large number of true-to-type planting material with high bioactive compound content of this valuable nutritional and medicinal species.

## 1. Introduction

Rhubarb is a widespread perennial plant (vegetable) that belongs to the Polygonaceae family which includes approx. sixty species of the *Rheum* genus [1], including *Rheum rhabarbarum*. Rhubarb plants are grown especially due to the easy and low input cultivation techniques that they require which make this plant profitable and commercially reasonable to grow [2]. Moreover, rhubarb plants have been known since ancient times due to their pharmacological potential. In general, the roots and rhizomes of *R. officinale* Baill., *R. palmatum* L. and *R. tanguticum* have been used and are still frequently used to treat constipation, inflammation and ulcers [3,4]. Furthermore, rhubarb contains a wide variety of bioactive compounds such as flavonoids, anthraquinone, glycosides, tannins, volatile oils and saponins [1,5,6,7], and plays an important role as an antifungal, antioxidant, hepatoprotective, nephroprotective and immune modulatory agent in pharmacological studies [8,9]. Various chemical constituents have been isolated from *Rheum* species, but the most important ones include stilbenes and their derivatives [10,11,12], flavonoids and phenolic acids [6,13,14,15,16,17], anthocyanins [1,7,18], and anthraquinone [19,20].

Anti-cancerous activities of the *Rheum* rhizomes have also been investigated successfully in human breast carcinoma (MDAMB-435S) and liver carcinoma cells (Hep3B) as reported by Rajkumar et al. [8,21]. Additionally, rhubarb species are also highly appreciated for their culinary values [1,22].

Thus, the continuously increasing demand for plants as raw material for drug preparation has become a serious problem due to the lack of planting material which could lead to the loss of plant populations or genetic diversity, degradation of natural habitat or species extinction [23].

Among the propagation methods of rhubarb plants, plant division is the most common method which guarantees homogenous offspring, but due to the low number of mother plants available, plant propagation and the extension of arable lands with rhubarb species resistant to adverse environmental conditions is almost impossible by plant division. Generative propagation of these species for cultivation is not recommended either, since rhubarb seeds are highly heterozygous and the new offspring are not morphologically homogeneous and stable [24]. In this regard, in vitro propagation could be a very effective alternative, since even a very small plant part, a few millimeters or even less, can be used as starting material for in vitro propagation which results in numerous clones in a much shorter time. Therefore, in vitro propagation of important medicinal plants and agricultural crops would become a reliable and necessary technique for high amount plant material production identical to mother plant [25,26].

Only a few scientific reports have been published until now [24,27,28] demonstrating that rhubarb can successfully be propagated in vitro with a probability of somaclonal variability occurrence in the micropropagated plants which infer the mandatory genetic fidelity testing of the mother plants and its offspring. In this context, it should be known that Sequence-Related Amplified Polymorphism (SRAP) markers present higher accuracy regarding genetic fidelity testing than Random Amplified Polymorphic DNA (RAPD) markers as described by Robarts and Wolfe in 2014 [29]. Therefore, the main aim of this research was to develop an efficient micropropagation protocol for local rhubarb planting material production. Another objective of our work was to check the genetic fidelity between in vitro plantlets and mother plants by using SRAP marker analysis. Furthermore, chemical analyses were also carried out to investigate the polyphenolic profile of the in vitro rhubarb plantlets and field grown plants, to confirm their pharmaceutical and economic importance. Since different factors can affect the chemical composition of *Rheum* species, such as genetics, botanical sources, geographical areas of production, harvest time and plant parts used, processing methods, this study aimed also to determine the biologically active constituents of rhizomes, petioles and leaves of the in vitro propagated *R. rhabarbarum* L. plants.

## 2. Results

### 2.1. In Vitro Culture Initiation and Stabilization

In the early stage of initiation (Figure 1a–c), it was observed that 20% of the explants were contaminated, 36% were necrotic and only 44% were viable.

In the in vitro culture stabilization stage, it was noticed that the explants transferred to the modified Murashige and Skoog medium (MSm) without plant growth regulators generated shoots that grew more in height (8–10 cm) and emerged roots, whereas on the media supplemented with Benzyladenine (BA) at concentrations of 0.5 mg L^−1^, 1 mg L^−1^, 2 mg L^−1^ and 4 mg L^−1^, the number of rhubarb plantlets obtained by axillary shoot proliferation increased proportionally with BA concentration (data not shown).

### 2.2. Multiplication Stage

The results show that the four types of cytokinins tested in the multiplication stage influenced the multiplication rate and axillary shoot development of rhubarb differently. The presence of BA or meta-topolin (mT) stimulated the proliferation of axillary shoots but limited petiole growth, whereas on the media supplemented with Kinetin (Kin) and 2-isopentenyladenine (2-IP), the multiplication rate was lower but petiole length was significantly greater (Table 1). Multiplication rates were calculated by dividing the number of shoots after the growth period by the number of shoots plated at the beginning of the experiment. The highest average multiplication rate (5 ± 0.5) was obtained with BA, whereas the biggest petiole length (9.6 ± 1.6 cm) was observed on media supplemented with 2-IP.

Apart from the MSm + 4 mg L^−1^ BA treatment (Figure 1d,e), rhizogenesis occurred in all the other treatments in the multiplication stage. The maximum number of roots obtained in the multiplication stage was 14.3 ± 3.7 roots/shoot clump in the experimental treatment with MSm + 4 mg L^−1^ Kin, where the percentage of rooted shoot clumps was also the highest (60%, Table 1).

Our results show that in the multiplication stage, cytokinins BA and mT provided the highest proliferation rates (5 ± 0.5 and 4.2 ± 0.5) while BA did not stimulate riyzogenesis and in the mT experimental treatment a small percentage of shoot clumps (17%) had an average number of roots of 3.0 ± 1.1. Based on the result obtained, we highly recommend the use of these two cytokinins in the multiplication stage of rhubarb plants.

### 2.3. Rooting and Acclimatization

Our results show that the rooting percentage was influenced by the number of shoots inoculated/vessel. As long as the number of shoots per vessel increased, the rooting percentage decreased. Thus, when 10 shoots/vessel were inoculated, the rooting percentage was 96%, while in the case of 20 shoots/vessel, the rooting percentage recorded was 86.3%.

In the *ex vitro* acclimatization stage in perlite, it was observed that the rhubarb plantlets obtained from all the treatments did not emerge roots in *ex vitro* conditions and all of them were affected by necrosis. The rhubarb plantlets rooted on hormone-free MSm media and those rooted in the multiplication stage on MSm + 4 mg L^−1^ Kin had high acclimatization percentages (90%), whereas those rooted in the multiplication stage on MSm + 4 mg L^−1^ 2iP had lower acclimatization percentages (80%). After four weeks of acclimatization, well-rooted and robust plants were developed; most of them with two leaves (Table 2).

The height of the acclimatized plants was lower than those before acclimatization. This can be explained by the fact that, before transferring them to perlite, 1–2 leaves per plantlet were removed and the petiole was also shortened to half length, to ensure a better insertion into perlite and to reduce leaf area to stimulate the rooting process. The survival rate of the transferred plants was 98% (Figure 1i,j) and after six months, small sized rhizomes (2–3 cm in length) were already developed (Figure 1k).

### 2.4. SRAP Analysis

To examine the genetic fidelity of in vitro propagated shoots, twelve SRAP primer combinations were screened for selection. Out of the twelve, only eight primer combinations (Table 3) generated clear and monomorphic electrophoretic profiles ranging from 184 to 1215 bp (Figure 2). Moreover, these results showed that the amplification patterns were identical between the mother plant and plants derived from the ninth in vitro subculture and demonstrated the efficiency of SRAP markers in genetic fidelity analysis of rhubarb.

### 2.5. Phenolic Compounds

A chromatogram of the lower-molecular-weight phenolic compounds is presented in Figure 3, indicating the presence of variable amounts of chemical constituents in rhubarb belonging to the class of polyphenols, depending on the analyzed plant organ. Separated peaks were obtained and identified by comparison of their Rt values with those of authentic standards, UV-VIS spectrum and similarity of more than 0.98, or co-chromatography with standards. The quantification was made using calibration curves obtained from different concentrations of standard solutions of the expected levels of each phenolic compound. Six raw material extracts were used for comparison; all of them being prepared from the same plant organs (rhizome, petiole and leaves) but from in vitro and field grown plants as well. The latter were also obtained from in vitro propagation and subsequently transferred to the field. In the analyzed plant extracts, 20 derivatives of gallic, protocatechuic, p-hydroxybenzoic, vanillic, chlorogenic, caffeic, syringic, p-coumaric and ferulic acid, rosmarinic acid and two derivatives, catechin, vitexin, rutin, resveratrol, isoquercitrin, quercitrin, apigenin and galangin were identified.

Among the 20 identified phenolic compounds, some of them are of a great importance for human health, such as: catechin, p-coumaric acid, rutin, caffeic and ferulic acid, rosmarinic acid and resveratrol. Gallic and p-coumaric acids were identified in different amounts in all extracts (field plants and in vitro grown plants). Regarding gallic acid, extracts from field plants contained higher amounts (32.6 μg g^−1^ in stalk extract to 148.5 μg g^−1^ in leaf extract) as compared to the in vitro plants extracts (28.3 μg g^−1^ in leaf extract to86.3 μg g^−1^ in rhizome extract). The same pattern was observed in p-coumaric acid; higher amounts being quantified in the extracts of field grown plants (1733.8 μg g^−1^ in rhizome extract, 426.7 μg g^−1^ stalk extract and 246.0 μg g^−1^ leaves extract) and lower amounts in in vitro plant extracts (Table 4). Caffeic acid was identified and quantified in the extracts of rhizome and leaves from both field and in vitro grown plants in similar amounts (23.9–58.2 μg g^−1^) (Table 4). Regarding catechin, our results show that rhubarb rhizome extracts were rich in catechin, but field grown plants had a higher amount of catechin (1463.3 μg g^−1^) compared to those grown in vitro (807.3 μg g^−1^).

The highest amount of rutin was determined in the leaves (672 µg g^−1^) of field grown and in vitro cultured plants (330.0 μg g^−1^). Apigenin and apigenin-8-C-glucoside (vitexin) were identified in the extracts in small amounts (11.6–93.6 μg g^−1^), higher quantities being observed in leaf extracts (114.3–515.6 μg g^−1^). Rosmarinic acid were quantified in significant amounts in all extracts, especially in rhizome (192.3–1506.5 μg g^−1^) and stalk extracts (118.6–179.0 μg g^−1^) along with another two derivatives of the acid which were quantified in very high amounts (6697.8 μg g^−1^) in rhizome extract (of field-grown plant). Significant amounts of resveratrol (229.4–371.7 μg g^−1^) were also found in the rhizome extract of both field-grown and in vitro propagated plants.

## 3. Discussion

In vitro propagation allows for a massive propagation of many plants, especially including endangered species but also the species in which propagation is limited or not satisfactory, or requires a lot of labor work, compromising its efficiency. Rhubarb plants are one of these species, in which propagation is less successful and unsatisfactory through common methods. Therefore, in vitro propagation of rhubarb plants is a potent alternative for mass plant material production to bridge the gap caused by the lack of planting material in agricultural, environmental and health issues.

In this study, despite the fact that the sterilization of the excised buds was carried out carefully, a high contamination percentage was observed in the initiation stage of the in vitro culture. The occurrence of this phenomenon can be explained by the fact that the buds used for in vitro culture initiation were taken from underground parts of the plants (rhizomes) which increase the risk of contamination as also reported by others [24]. These results suggest that a more severe sterilization process of the excised plant fragments is needed for a successful in vitro culture of rhubarb.

Plant growth regulators, when added to in vitro culture media, have various effects on proliferation and plant development as revealed by our study also. The cytokinins added to the culture media had different effects on shoot development and proliferation rate of the rhubarb in vitro plantlets. Among the different cytokinins used in the same concentrations (4 mg L^−1^), maximum values of proliferation were obtained on the medium MS supplemented with BA and mT, while other cytokinins such as 2iP and Kin stimulated the shoot length and their rooting more. Similar observations regarding the superiority of BA and the effects of other cyokinins were also reported by Kozak and Salata [27] in *R. rhaponticum.*

Lepse [24], in one of his studies carried out on a local clone of *R. rhabarbarum* din Latvia, reported a proliferation rate of 1.7 on MS medium supplemented with BAP 1 mg L^−1^ + IBA 1 mg L^−1^ and another of 2.6 obtained on MS medium supplemented with BAP 1.5 mg L^−1^ + IBA 0.7 mg L^−1^. In this study, it was shown that a concentration of 4 mg L^−1^ BA added to the MS medium without adding auxins led to a proliferation rate of 5 in *R. rhubarbarrum*. These results indicate that the use of cytokinins alone in high concentrations in the culture media stimulates the proliferation rate more than the combination of cytokinins and auxins.

In vitro root regeneration was observed in all the culture media tested, except MSm + 4 mg L^−1^ BA. Superior values of in vitro rooting were obtained on the media supplemented with Kin (60%) and 2iP (31%), acting as rooting stimulators. Moreover, our findings also suggest that the number of explants used per culture vessel influenced in vitro rooting on hormone-free MSm medium; the rooting percentage was much higher when 10 shoots were inoculated per vessel as compared to those of 20 shoots/vessel. In the scientific literature, the information is scant regarding in vitro and ex vitro rooting of *R. rhabarbarum.* Lepse [24] declares, in one of his studies, that successful rhizogenesis was induced using MS medium containing 0.1 mg L^−1^ IBA, but in contrast, in *R. rhaponticum,* a 100% in vitro rooting percentage was reported in MS medium without PGRs (BA, Kinetin, 2-iP or thidiazuron) added followed by an *ex vitro* acclimation percentage of 100% also [27].

Successful procedures for in vitro propagation of *Rheum rhabarbarum* and *R. raponticum* L. plants have already been reported [24,27,28]. However, these reports also highlighted the risk of somaclonal variability occurrence in the micropropagated plants. Genetic variations induced by somaclonal variations in tissue cultured plants are beneficial for crop improvement or plant breeding processes, but somaclonal variations which occur in the micropropagation process makes it mandatory to test the genetic fidelity of the plants [30].

In this study, the in vitro propagated rhubarb offspring were subjected to genetic fidelity analyses to check the true-to-type nature of the new offspring to the mother plants. To test the genetic fidelity of the plants, the SRAP marker system was applied. Our results are in accordance with those reported by Sun et al. (2014) [31] and Li et al. (2014) [32] concerning the suitability and effectiveness of SRAP and PCR molecular marker systems for genetic fidelity assessment of tissue cultured plants. A comparison of the number of polymorphic bands and their ranges with the mother plant clearly revealed the true-to-type clones. The visual observations of the micropropagated plants did not show any variations among the plants.

The chemical composition of the in vitro rhubarb plants is still poorly investigated. The phenolics might be classified into six groups: anthraquinones, anthrones, phenylbutanones, acylglucoses, galloylglucoses and phenolic acids [33]. Anthraquinones and their glycosides are the most common phytochemicals in *Rheum* genus [12,20,34]. These compounds are known for their laxative properties and anti-fungal activity [12]. Free and glycosylated phenolic compounds from the extracts of this genus are not sufficiently investigated, which could draw the attention of any researcher and serve as a topic of further research.

Gallic acid was found in rhizome of *R. palmatum*, *R. undulatum* and *R. rhaponticum* and different *Rheum* species from Japan [14,33]. Similar results have been reported by Medyńska and Smolarz (2005) [14] using a simple extraction method with methanol for the determination of p-coumaric acid from *R. palmatum*, *R. rhaponticum* and *R. undulatum*. These results are in agreement with our results regarding the quantity of phenolic acids in the underground part of the tissue cultured rhubarb plants.

Caffeic acid is an organic compound and a hydroxycinnamic acid with high antioxidant properties, immunomodulatory and anti-inflammatory activity [35,36,37] which was detected in low amounts in the rhizomes and leaves of both in vitro and field-grown plants but not in the stalk of the rhubarb plants.

Catechin (a flavan-3-ole), found especially in green tea leaves, grape seeds and cocoa beans, has several health properties including antioxidant or chemo-preventive activities [38]. Similar catechin contents were also found in 52 different rhubarb specimens examined from Japan ranging from 4.04 ± 1.12 to 8.22 ± 1.24 mg/g [33].

Rutin (quercetin-3-O-rutinoside) is a flavonol synthesized through the phenylpropanoid metabolic pathway, abundantly found in plants, a vital nutritional component of food [39] with different bioactive properties, including antioxidant, cytoprotective, neuroprotective and cardioprotective properties, as well as anticarcinogenic activity [40,41]. Our results show that this flavonol was found only in the rhubarb leaves and not in other organs. Previous reports show that in calli cultures, a high accumulation of rutin was influenced by phenyl alanine and salicylic acid which were added to the culture media [42].

Furthermore, the group of secondary metabolites called “caffeic acid derivatives” include rosmarinic acid and its derivatives. Rosmarinic acid and its derivatives are plant-based compounds produced through shikimate/phenylpropanoid biosynthetic pathway, found in a wide variety of plants with antioxidant, anti-inflammatory and anti-diabetic effects [43,44,45]. It is of a great importance that rosmarinic acid is easily produced in cultured plant cells. As our results suggest, the highest content of rosmarinic acid was found in the underground part of the field grown rhubarb plants. Rosmarinic acid has five key structural groups in the molecule [46], and it is not found alone, most of the time being accompanied by its derivatives in plants.

Further studies are currently being carried out for the exact identification of these compounds of such a great importance for nutritional and bioactive activity. All the bioactive properties of rosmarinic acid and the high quantities found in rhubarb extracts make this plant more valuable in terms of medical uses.

Resveratrol is a phytoalexin whose synthesis in plants is induced in response to different stressors [47,48]. It is known for its antioxidant and anticancer properties, and it is also a protective of cardiovascular system. It plays an important role in insulin lower “bad” cholesterol level regulation [49]. The richest source of resveratrol is *Polygonum cuspidatum* roots (*Polygonaceae* family), which belongs to the same botanical family as *R. rabarbarum*. Few plants are known to have resveratrol in their chemical composition [48,49,50], thus, every newly discovered source is very important for nutritional and medical purposes.

Quantifying phenolic compounds in the extracts of different plant parts developed under different growing conditions, suggests that, in general, plants grown in field conditions had higher amounts of phenolics accumulated in different plant parts than those grown In vitro. However, the rhizomes and stalks of the in vitro plant extracts had low quantities of vanillic acid, while no vanillic acid was detected in the field grown plant extracts.

## 4. Materials and Methods

### 4.1. In Vitro Culture

#### 4.1.1. Initiation and Stabilization

To establish the tissue culture, rhizomes were harvested at the end of March 2016 from an open, agricultural field from Sighișoara (Mureș county, Romania, 46°13′1″ N, 24°47′28″ E). The axillary buds from the collected rhizomes were excised and washed thoroughly first with running tap water for 10 min to remove excess dirt. Then, the excised buds were subjected to further washing with tap water using a magnetic stirrer to eliminate all the dust and impurities followed by decontamination with bleach solution of 20% (ACE-Procter and Gamble, Romania; <5% active ingredient) for 20 min and several rinses with sterile distilled water.

The Murashige and Skoog medium (MS) [51] was modified (MSm) to culture the aseptic axillary buds. The MSm medium was prepared from stock solutions containing macro- and micro-nutrients, Myo-inositol100 mg L^−1^, vitamins, thiamine-HCl 1 mg L^−1^, pyridoxine-HCl 0.5 mg L^−1^, nicotinic acid 0.5 mg L^−1^, and 30 g L^−1^ sucrose as carbon source which was added separately. The pH of the medium was adjusted to 5.8 with 0.1 N NaOH and/or 0.1 N HCl before Plant Agar (4 g L^−1^ Duchefa Biochemie BV, the Netherlands) was added to solidify the culture media. Inoculation was made in a laminar flow hood. The 25 excised buds of *R. rhabarbarum* L. were inoculated on a hormone free MSm medium in 11.5 cm tall glass test tubes (ø 2 cm) containing 5 mL medium.

The regenerated axillary shoots from axillary buds were further multiplied at two-month intervals through several passages on a growth regulator-free MSm medium and simultaneously on other MSm media supplemented with 6-Benzyladenine (BA) in different concentrations (0.5 mg L^−1^; 1 mg L^−1^; 2 mg L^−1^ and 4 mg L^−1^) for culture stabilization.

#### 4.1.2. Multiplication Stage

In this stage, MSm gelled with 4 g L^−1^ Plant Agar was used as basal medium, supplemented with 4 different plant growth regulators (4 mg L^−1^ each) as follow: BA, kinetin (Kin), 2-isopentenyladenine (2-iP) and meta-topolin (mT). In each vessel (720 mL glass jar, 9 cm diameter, 13.5 cm high) with screw cap and ventilation holes (4 mm) 100 mL culture medium was dispensed. The explants used were obtained from the hormone-free MSm. Before inoculation to the new medium, the already developed roots and leaves of the plantlets were removed, and the length of the petiole was also shortened to the half of its length. Five rhubarb plantlets were inoculated per culture vessel in ten vessels/treatment. The in vitro cultures were maintained at 16/8 h light/dark photoperiod at 32.4 μmol·m^−2^s^−1^ light intensity, 23 ± 3 °C and 50% humidity in a growth room. After two months of culturing, five vessels from each experimental treatment were randomly selected and the number of rhubarb plantlets was recorded. The petiole length and multiplication rate were also measured and calculated.

#### 4.1.3. In Vitro Rooting and Acclimatization

Two-months old rhubarb in vitro plantlets obtained on MSm medium supplemented with 4 mg L^−1^ BA were used to test the rooting capacity of the plantlets. For root formation, the shoot clumps developed from the individual explants were separated. The leaves of each plantlet were removed, and the stalk shortened to half of its length. The shoots were then introduced in the growth regulator-free MSm medium. Two different experimental treatments were established: with 10 and 20 shoots per vessel, with three repetitions each with five culture vessels per repetition. After four weeks, both rooted and non-rooted plantlets were counted, and the rooting percentages were calculated. Plant height, leaf number/plantlet and root length were also recorded from five randomly sampled plantlets from each culture vessel. Then, 40 rooted and 40 non-rooted plantlets were equidistantly spaced into the rooting substrate prepared of 3 L of perlite (2 mm granularity) and 3 L of tap water in square shaped, 5-L plastic tubs. After five weeks, the rooted rhubarb plantlets were transplanted into cell trays (ELFORM EPE45 trays from Horticola, Oradea, BH, Romania) containing peat-based potting mix (Klassman TS3, Klasmann-Deilmann Gmbh, Geeste, Germany) and kept in greenhouse conditions (21 ± 4 °C, under natural photoperiod conditions). One month later, plant height, leaf number/plant, root length and rooting percentage were measured and calculated.

### 4.2. SRAP Analysis

Total genomic DNA was isolated from the mother plant and nine randomly selected in vitro propagated plants from the 9th subculture. The DNA was isolated using the CTAB-based method as published by Lodhi et al. (1994) [52] and improved by Pop et al. (2003) [53]. DNA purity and concentration were determined with NanoDrop-1000 spectrophotometer (Thermo Fisher Scientific, USA). In order to perform SRAP analysis, 12 primer combinations (Table 5) were screened, followed by genotyping with the eight most representative primer combinations (me1/em3; me2/em3; me3/em6; me4/em2; me6/em8; me8/em2; me5/em2; me6/em1) to assess the genetic fidelity between the micropropagated plants and the mother plant.

PCR reactions were carried out according to Li and Quiros (2001) [54] protocol, adjusting the reaction volumes to 15 μL containing 3 µl of 5 × Green Go Taq flexi buffer, 1.5 mM MgCl2, 0.2 mM of dNTPs, 0.3 µM of both forward and reverse primer, 1 U of Taq DNA polymerase, (all purchased from Promega Madison WI, USA) and 2 μL of sample DNA (50 ng/μL). The PCR program included the following cycles: initial denaturation at 94 °C for 5 min, followed by 5 cycles of 94 °C for 1 min, 35 °C for 1 min, 72 °C for 1 min, and 35 cycles of 94 °C for 1 min, 50 °C for 1 min, 72 °C 1 min, with final elongation at of 72 °C for 10 min. Separation of the amplified products was performed by electrophoresis on 2% agarose gel with ethidium bromide detection. The electrophoretic profiles were visualized under UV and the images were captured using BioSpectrum AC Imaging System.

### 4.3. Chemical Analyses

The phenolic spectra of the plant material, both from in vitro propagated and field grown plants were determined by High-performance liquid chromatography (HPLC) analysis.

#### 4.3.1. Chemicals and Standards

Standard substances of phenolic acids: gallic (3,4,5-trihydroxybenzoic), protocatechuic (3,4-dihydroxybenzoic), p-hydroxybenzoic (4-hydroxybenzoic), vanillic (4-hydroxy-3-methoxybenzoic), caffeic (3,4-dihydroxy-cinnamic), syringic (4-hydroxy-3,5-dimethoxybenzoic), p-coumaric (4-hydroxycinnamic), ferulic (3-methoxy-4-hydroxycinnamic), sinapic (3,5-dimethoxy-4-hydroxycinnamic) and rosmarinic acid were purchased from Sigma-Aldrich (St. Louis, MO, USA), as well as catechin, rutin, vitexin, resveratrol, isoquercitrin, quercitrin, apigenin and galangin. The solvents used for chromatography were acetonitrile and methanol of HPLC ultra-gradient grade, supplied by Merck (Darmstadt, Germany). Millipore water was obtained from a Milli-Q Plus water system.

#### 4.3.2. Plant Material and Extraction

For chemical constituent determination, the plant material (rhizomes, petioles and leaves) was obtained from both micropropagated plants cultured on MSm + 4 mg L^−1^ BA medium (9th subculture) and young, field grown plants resulted from previous in vitro propagation processes. Different amounts of dried plant material were ground into fine powder, weighted (0.15–0.75 g) and subjected to extraction. All the constituents have been extracted with acidified methanol (2 × 20 mL) shaken for 24 h and followed by ultra-sonication for 15 min. Filtered supernatants were concentrated to dryness under reduced pressure using a Büchi evaporator. Samples containing phenolic molecules were dissolved in 2 mL HPLC pure methanol, filtered through Millipore filter (0.45 µm) and injected in the HPLC system.

#### 4.3.3. Analytical HPLC-PDA Determination

HPLC analysis was carried out using a Shimadzu binary gradient unit LC-10AD VP (Shimadzu Instruments, Kyoto, Japan) system, equipped with a SCL10AVP system controller, SPP-M20A Prominence Diode Array Detector, LC-10ADSP binary pumps, CTO-10AVP column oven and SIL-10AF autosampler [55]. Separation was performed on a Teknokroma Mediterranean Sea 18 15 × 0.46 cm, i.d. 5 µm column, maintaining a flow rate of 1 mL/min. The gradient consisted of 2.5 pH (adjusted with orthophosphoric acid) water (solvent A) and acetonitrile (solvent B). Starting gradient was 5% solvent B, changed to 9% B in 3 min (min 12 of separation); min 20, 13% B; min 30, 33% B; min 42 to 60 liniar gradient to 43% B; min 65, 90% B; min 70, 100% B, than decreasing to 5% B until min 78. Column temperature was maintained to 24 °C (CTO-10AVP column oven, Shimadzu Corporation, Kyoto, Japan), and the separation was monitored at 220–400 nm [56]. Commercial standards of phenolic acids: gallic, p-OH-benzoic, chlorogenic, caffeic, siringic, ferulic and rosmarinic acid, flavonoids (catechin, quercetin, rutin, hyperosid, quercitrin, miricitrin, isoquercitrin, luteolin, apigenin and kaempherol) were used as reference compounds.

### 4.4. Statistical Analysis

The analysis of variance performed to analyze the data and compare the means of every treatment according to Tukey’s HSD (honestly significant difference) test (*p* < 0.05). The values shown are means ± S.E. SRAP gel images were analyzed using TotalLab TL120 software (Nonlinear Dynamics, Newcastle upon Tyne, UK) to determine the number and size of the amplification products. Pearson’s correlation was also carried out to investigate the relationships between the chemical compounds in the rhizomes, stalks and leaves from both in vitro and field plants.

## 5. Conclusions

The micropropagation protocol for *R. rhabarbarum* L. developed in this study can be an efficient tool to propagate this multipurpose industrial crop on a large scale. The DNA marker system used proved the true-to-type nature of the in vitro plantlets to the mother plants, enhancing the potential of this propagation method. Moreover, six hydro-alcoholic extracts of rhizome, stalk and leaves of *R. rhabarbarum* L. were investigated for their content in low-molecular-polyphenolic substances. Twenty phenolic acids and flavonoids were isolated and identified as active constituents in the extracts. The major constituents isolated from in vitro cultivated material were: catechin, p-coumaric acid, isoquercitrin, resveratrol and rosmarinic acid and its derivatives in rhizome, and rutin in leaves. Extracts from plant material were rich in catechin, p-coumaric acid, ferulic acid, resveratrol and rosmarinic acid and its derivatives (rhizome extract), and p-coumaric and rosmarinic acid (stalk extract). Therefore, our findings suggest that in vitro propagation of *R. rhabarbarum* L. could be considered as an alternative to provide important source of bioactive phytochemicals, which could be used further in different pharmacological studies. Moreover, a large quantity of high-quality planting material can also be obtained even, for agricultural purposes.

## Figures and Tables

**Figure 1 plants-09-00656-f001:**
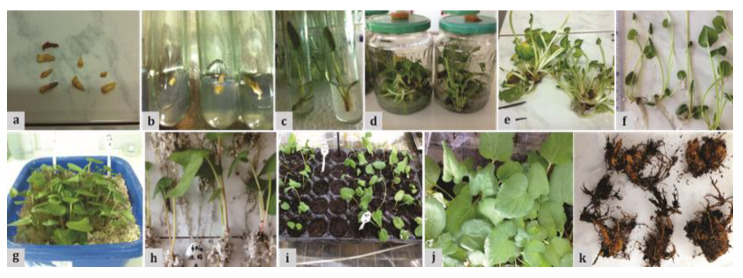
In vitro propagation of *Rheum rhabarbarum* L.: (**a**–**c**) In vitro culture initiation; (**d**,**e**) Multiplication stage on the medium MSm + 4 mg L^−1^ BA; (**f**) in vitro rooted plantlets on hormone-free MSm; (**g**–**j**) Acclimatization stage; (**k**) Tissue cultured rhubarb rhizomes.

**Figure 2 plants-09-00656-f002:**
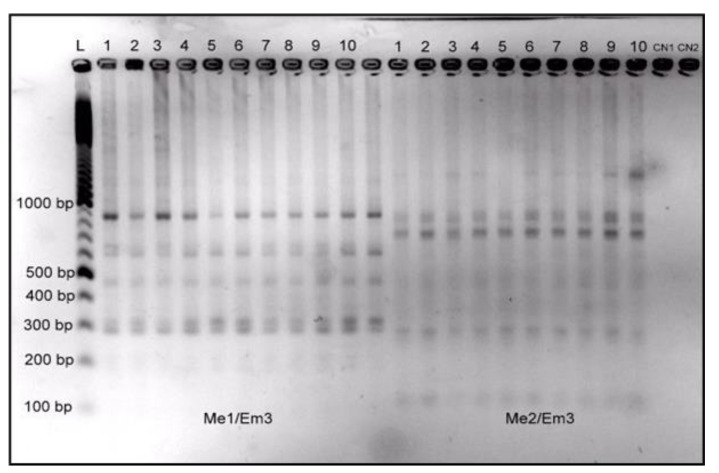
Electrophoretic SRAP profiles of rhubarb plants. (**1**) mother plants and (**2–10**) nine tissue cultured plants after the ninth in vitro culture. (**L**) ladder; (**CN**) negative controls.

**Figure 3 plants-09-00656-f003:**
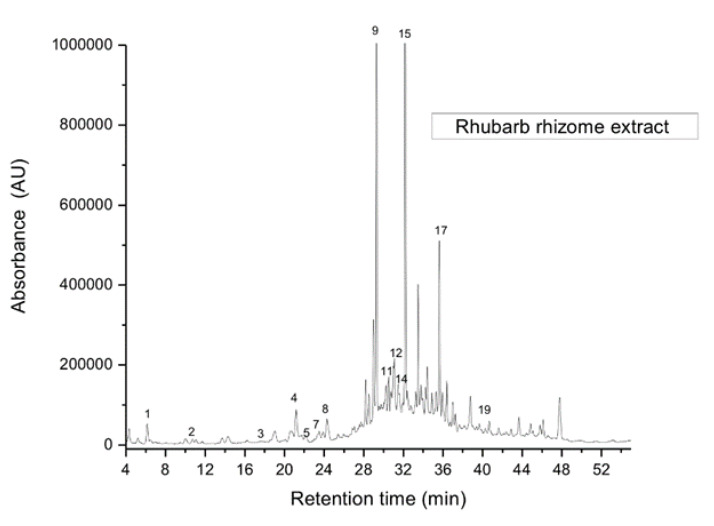
Chromatogram of identified polyphenolic compounds detected in extracts of rhubarb rhizome. Peak identification, see Table 4.

**Table 1 plants-09-00656-t001:** The influence of cytokinins upon in vitro multiplication in *Rheum rhabarbarum* L. after eight weeks.

Types of Hormone Used (4 mg L^−1^)	Average No. of Shoot Clumps/ Vessel	Average Petiole Length (cm)	Rooted Clumps (%)	Average No. of Roots/ Shoot Clump	Multiplication Rate/Explant
BA	24.5 ± 2.5 ^c^	3.7 ± 0.3 ^a^	0	0.0 ± 0.0 ^a^	5 ± 0.5 ^c^
Kin	12.5 ± 1.1 ^a^	8.00 ± 0.3 ^c^	60	14.3 ± 3.7 ^d^	2.5 ± 0.2 ^a^
mT	21.2 ± 2.3 ^b^	5.7 ± 0.2 ^b^	17	3.0 ± 1.1 ^b^	4.2 ± 0.5 ^b^
2-IP	12.3 ± 1.5 ^a^	9.6 ± 1.6 ^d^	31	5.8 ± 2.4 ^c^	2.6 ± 0.5 ^a,1^

^1^ The values shown are means ± SE. Different lowercase letters indicate significant differences between the means of different treatments according to Tukey’s HSD test (*p* ≤ 0.05).

**Table 2 plants-09-00656-t002:** Biometrical measurements in rhubarb plants one month after acclimatization.

Types of Hormone Used (4 mg L^−1^)	Plant Height (cm)	No. of Leaves/Plant	Root Length (cm)
2iP	5.2 ± 0.6 ^c^	2.2 ± 0.2 ^a^	11.1 ± 1.3 ^b^
Kin	3.9 ± 0.6 ^a^	2.1 ± 0.3 ^a^	10.0 ± 1.5 ^a^
only MSm	4.3 ± 0.6 ^b^	2.0 ± 0.2 ^a^	10.1 ± 1.4 ^a,1^

^1^ The values shown are means ± SE. Different lowercase letters indicate significant differences between the means of different treatments according to Tukey’s HSD test (*p* ≤ 0.05).

**Table 3 plants-09-00656-t003:** Amplification products resulted with sequence-related amplified polymorphism (SRAP) primers of *Rheum rhabarbarum* L.

SRAP Primer Combination	Number of Monomorphic Bands	Range of Amplification Products (bp)
me1/em3	4	289–753
me2/em3	6	184–975
me3/em6	5	203–1215
me4/em2	3	294–548
me6/em8	3	220–645
me8/em2	5	287–756
me5/em2	3	356–793
me6/em1	4	278–654

**Table 4 plants-09-00656-t004:** Phenolic composition of in vitro and field grown *Rheum* plant extracts. (μg g^−1^ dry material).

	Compound	In vitro			Field		
	Type of Extract	Rhizome	Stalk	Leaf	Rhizome	Stalk	Leaf
1	Gallic acid	86.3 ^b^	33.9 ^a^	28.3 ^a^	124.0 ^b,^*	32.6 ^a^	148.5 ^b,^*
2	Protocatechuic acid	4.6 ^a^	2.3 ^a^	- ^a^	57.7 ^c,^*	33.7 ^b,^*	21.5 ^a,^*
3	p-OH-benzoic acid	14.5 ^b^	19.7 ^c,^*	- ^a^	- ^a^	12.4 ^b^	12.2 ^b,^*
4	Catechin	807.3 ^b^	- ^a^	- ^a^	1463.3 ^b,^*	- ^a^	- ^a^
5	Vanillic acid	25.3 ^c,^*	15.3 ^b,^*	- ^a^	-	-	-
6	Clorogenic acid	- ^a^	7.7 ^b^	- ^a^	- ^a^	6.1 ^b^	- ^a^
7	Caffeic acid	23.9 ^b^	- ^a^	41.1 ^c,^*	58.2 ^c,^*	- ^a^	33.3 ^b^
8	Syringic acid	18.3 ^b^	16.2 ^b^	- ^a^	71.9 ^b,^*	24.5 ^a,^*	82.7 ^b,^*
9	p-cumaric acid	624.1 ^c^	44.2 ^a^	80.4 ^b^	1733.8 ^b,^*	426.7 ^a,^*	246.0 ^a,^*
10	Vitexin	- ^a^	- ^a^	93.6 ^b^	- ^a^	- ^a^	515.6 ^b,^*
11	Rutin	- ^a^	- ^a^	330.0 ^b^	- ^a^	- ^a^	672.0 ^b,^*
12	Ferulic acid	69.2 ^c^	57.1 ^b^	- ^a^	2690.3 ^c,^*	71.4 ^b,^*	- ^a^
13	Isoquercitrin	239.5 ^c,^*	147.2 ^a^	174.2 ^b^	- ^a^	218.0 ^b,^*	572.0 ^c,^*
14	RA derivate_1_	260.9 ^c^	55.9 ^b^	- ^a^	1318.6 ^c,^*	158.9 ^b,^*	54.7 ^a,^*
15	RA derivate_2_	17281.3 ^c,^*	179.0 ^b^	- ^a^	6697.8 ^c^	1072.4 ^b,^*	506.7 ^a,^*
16	Rosmarinic acid (RA)	192.3 ^b^	179.0 ^b,^*	- ^a^	1506.5 ^c,^*	118.6 ^b^	65.8 ^a,^*
17	Resveratrol	371.7 ^b,^*	- ^a^	- ^a^	229.4 ^b^	- ^a^	- ^a^
18	Quercitrin	- ^a^	- ^a^	54.9 ^b^	- ^a^	- ^a^	206.5 ^b,^*
19	Apigenin	11.6 ^b^	- ^a^	56.9 ^c^	78.7 ^a,^*	122.0 ^b,^*	114.3 ^b,^*
20	Galangin	- ^a^	3.5 ^b,^*	6.3 ^c^	3.5 ^a,^*	- ^a^	245.3 ^b,^*^,1^

^1^ The values shown are means. Different lowercase letters indicate significant differences between the means of chemical compounds in different type of extracts both from in vitro and field grown plants according to Tukey’s HSD test (*p* ≤ 0.05). * indicate significant differences between the means of in vitro and field grown plants of the same type of extract.

**Table 5 plants-09-00656-t005:** SRAP primer combinations used in this study.

No.	Forward Primer	Sequences 5′-3′	Reverse Primer	Sequences 3′-5′
1	Me1	TGA GTC CAA ACC GGA TA	Em6	GAC TGC GTA CGA ATT GCA
2	Me2	TGA GTC CAA ACC GGA GC	Em1	GAC TGC GTA CGA ATT AAT
3	Me2	TGA GTC CAA ACC GGA GC	Em6	GAC TGC GTA CGA ATT GCA
4	Me3	TGA GTC CAA ACC GGA AT	Em3	GAC TGC GTA CGA ATT GAC
5	Me4	TGA GTC CAA ACC GGA CC	Em2	GAC TGC GTA CGA ATT TGC
6	Me5	TGA GTC CAA ACC GGA AG	Em2	GAC TGC GTA CGA ATT TGC
7	Me5	TGA GTC CAA ACC GGA AG	Em6	GAC TGC GTA CGA ATT GCA
8	Me6	TGA GTC CAA ACC GGA CA	Em1	GAC TGC GTA CGA ATT AAT
9	Me6	TGA GTC CAA ACC GGA CT	Em8	GAC TGC GTA CGA ATT CAC
10	Me8	TGA GTC CAA ACC GGA CT	Em2	GAC TGC GTA CGA ATT TGC
11	Me8	TGA GTC CAA ACC GGA CT	Em3	GAC TGC GTA CGA ATT GAC
12	Me8	TGA GTC CAA ACC GGA CT	Em6	GAC TGC GTA CGA ATT GCA

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
