# Peer review of "Micropropagation, Genetic Fidelity and Phenolic Compound Production of *Rheum rhabarbarum* L."

_plants, 2020, doi:10.3390/plants9050656_

Round 1

Reviewer 1 Report

The authors presented for publication very well prepared and very well written manuscript.

They provided valuable protocol for micropropagation of the medicinal plant Rheum rhabarbarum (rhubarb) that is appreciated for its content of wide variety of bioactive compounds with pharmacological potential. The propagation of plants by standard method of division the plant sis not efficient because of the lack of mother plants. The micropropagation has the potential to fulfil the demands of the market. There is only a few scientific reports showing the successful propagation of rhubarb in vitro. The authors optimized the protocol and proved the genetic fidelity of offsprings using the more reliable method of Sequence-related amplified polymorphism markers. They isolated and identified 20phenolic acids and flavonoids from Rhizome, leaves and petioles of micropropagated plants and compared the contents with extracts of plants cultivated in the field. Even though the contents of bioactive compounds were higher in the plants grown in the field, the micropropagation of this species can provide an efficient tool for further pharmacological studies. Therefore, their work is original and valuable. As I have already mentioned in the comments in the form, I would recommend to write the expression „in vitro“ in Italics and to correct some probably only typing errors. That is what I have meant by the „minor revisions“.

The manuscript is well prepared for publication and I would propose to accept it after very minor revisions of mainly typing errors.

line 107 - rhyzogenesis

line 108 - rootsof

line 311 - kinetine x kinetin

line 312 - topoline x topolin

line 361 - p-hydroxibenzoic (4-hydroxibenzoic)

I would recommend to write "in vitro" in Italics, as it is common in scientific articles and it will contribute to the comprehension of the text. 

Reviewer 2 Report

The manuscript is very interesting as it contains some useful information on the micropropagation of Rheum rhabarbarum, valuable medicinal plant. The Authors have achieved high percentages of proliferation, rooting and acclimatization. Although the paper is based on in vitro culture of rhubarb, the outcomes can be useful also when working with the micropropagation of other species. The genetic fidelity of micropropagated plants was tested from the 9th subculture.

A very interesting aspect of the research is the comparison of the phenolic profile of the micropropagated plants and the young plants grown in the field, comparing different vegetal organs (rhizome, stalk, leaf). In this regard, I suggest to the Authors to provide, if available, some information relating to the bio-compounds produced by the mother plant from which they obtained the initial explants to start in vitro culture. In this way, it will be possible to verify whether micropropagation altered the biochemical profile.

To be ready for publication on Plants, the manuscript needs some minor revisions; it contains a good “Introduction”, “Material and Methods” and “Results” are clearly described, the “Discussion” part is equilibrated and pertinent with the outcomes of the investigation. Some minor corrections and suggestions to improve even more the manuscript are directly reported on the text with comments which I ask to the Authors to take into consideration.
